# Peritoneal Fluid Cytokines Reveal New Insights of Endometriosis Subphenotypes

**DOI:** 10.3390/ijms21103515

**Published:** 2020-05-15

**Authors:** Jieliang Zhou, Bernard Su Min Chern, Peter Barton-Smith, Jessie Wai Leng Phoon, Tse Yeun Tan, Veronique Viardot-Foucault, Chee Wai Ku, Heng Hao Tan, Jerry Kok Yen Chan, Yie Hou Lee

**Affiliations:** 1Translational ‘Omics and Biomarkers group, KK Research Centre, KK Women’s and Children’s Hospital, Singapore 229899, Singapore; zhou.jieliang@kkh.com.sg; 2Division of Obstetrics and Gynaecology, KK Women’s and Children’s Hospital, Singapore 229899, Singapore; bernard.chern.s.m@singhealth.com.sg (B.S.M.C.); gmskcw@nus.edu.sg (C.W.K.); 3OBGYN-Academic Clinical Program, Duke-NUS Medical School, Singapore 169857, Singapore; jessie.phoon.w.l@singhealth.com.sg (J.W.L.P.); tan.heng.hao@singhealth.com.sg (H.H.T.); jerry.chan.k.y@singhealth.com.sg (J.K.Y.C.); 4Department of Obstetrics and Gynaecology, Singapore General Hospital, Singapore 169608, Singapore; pcbartonsmith@gmail.com; 5Department of Reproductive Medicine, KKH, Singapore 229899, Singapore; tan.tse.yeun@singhealth.com.sg (T.Y.T.); veronique.viardot@singhealth.com.sg (V.V.-F.)

**Keywords:** endometriosis, cytokines, peritoneal fluid, microenvironment, precision medicine

## Abstract

Endometriosis is a common inflammatory gynecological disorder which causes pelvic scarring, pain, and infertility, characterized by the implantation of endometrial-like lesions outside the uterus. The peritoneum, ovaries, and deep soft tissues are the commonly involved sites, and endometriotic lesions can be classified into three subphenotypes: superficial peritoneal endometriosis (PE), ovarian endometrioma (OE), and deep infiltrating endometriosis (DIE). In 132 women diagnosed laparoscopically with and without endometriosis (*n* = 73, 59 respectively), and stratified into PE, OE, and DIE, peritoneal fluids (PF) were characterized for 48 cytokines by using multiplex immunoassays. Partial-least-squares-regression analysis revealed distinct subphenotype cytokine signatures—a six-cytokine signature distinguishing PE from OE, a seven-cytokine signature distinguishing OE from DIE, and a six-cytokine-signature distinguishing PE from DIE—each associated with different patterns of biological processes, signaling events, and immunology. These signatures describe endometriosis better than disease stages (*p* < 0.0001). Pathway analysis revealed the association of ERK1 and 2, AKT, MAPK, and STAT4 linked to angiogenesis, cell proliferation, migration, and inflammation in the subphenotypes. These data shed new insights on the pathophysiology of endometriosis subphenotypes, with the potential to exploit the cytokine signatures to stratify endometriosis patients for targeted therapies and biomarker discovery.

## 1. Introduction

Endometriosis inflicts 6–10% of women of reproductive age, with affected women suffering from debilitating pelvic pain, dysmenorrhea, dyspareunia, and painful defecation. The prevalence of subfertility or infertility is higher in the endometriosis population, with up to half of endometriotic women suffering from reduced fecundity [1]. Endometriotic lesions can be classified into three subphenotypes: superficial peritoneal endometriosis (PE), ovarian cysts (ovarian endometrioma; OE), and deep infiltrating endometriosis (DIE) [2]. Superficial peritoneal lesions refer to implants found on the pelvic organs or pelvic peritoneum. They are typically white, red, or blue–black power burns and may be phenotypically progressive over time [3,4,5]. Endometriomas are formed when endometrial tissue implants on the ovary surface or penetrate into the ovary [6,7,8]. By contrast, DIE involves nodular lesions which invade the surrounding organs beneath the peritoneum [9]. They are commonly found on the uterosacral ligaments, bladder, vagina, and intestine and are more aggressive in nature [10]. The endometriosis subphenotypes are being recognized as clinicopathologically different from one another and are thought to be separate entities [3,10,11].

Many theories and hypotheses have been proposed to explain the pathogenesis and development of endometriosis [12,13]. The most widely accepted theory to explain the etiology of endometriosis is the retrograde menstrual hypothesis. During menses, viable menstrual fragments of eutopic endometrial origin back-traverse through the fallopian tubes, into the peritoneal cavity, which subsequently leads to these cells being implanted onto extra-uterine ectopic sites. However, this theory is insufficient to explain the presence of the disease in early puberty and in adolescents [14], and it does not address why endometriosis only occurs in a certain subset of the female population. Other theories have been put forth, including the coelomic metaplasia theory, endometrial stem/progenitor cells theory, and Müllerianosis theory. These complementary theories and hypotheses help to generally frame our understanding of plausible mechanisms of the endometriosis pathogenesis and allow for the testing of hypotheses and improvement of the currently existing treatment options and the introduction of new treatments. Recent advances had been made with respect to elucidating partial progestin response based on the progesterone receptor status of endometriosis patients [15], which infers the prospect of precision or targeted medicine for treating endometriosis. In addition, key non-hormonal inflammation or signaling pathways associated with endometriosis subphenotypes could be exploited for therapeutic advantage, particularly in the era of targeted treatment and precision medicine [16].

Ectopic endometriotic lesions secrete chemokines into their surrounding tissues, recruiting immune cells, which in turn secrete cytokines and growth factors, such as TNF-α, IL-1β, IL-6, IL-8, IL-17, EGF, FGF, and VEGF, creating locally specific microenvironments that develop a reciprocal interaction with the peritoneal fluid (PF) [17,18,19]. The interaction of cytokines constitutes molecular signatures that are used to orchestrate immune and other complex responses in endometriosis development, plausibly in the evolution of lesion subphenotypes [14,20]. With increased appreciation that immunomodulatory proteins are part of an overall signaling network [21,22], significant discoveries of pro-angiogenic, macrophage-derived, and endometriosis-associated infertility cytokine signatures have been made in recent years [23,24,25,26,27]. While these studies provide insight into major pathways involved in disease stages, several outstanding gaps remain in terms of the cytokine relationship with endometriosis subphenotypes [28] and their implications in stratified treatment. Here, in this study, we investigated PF cytokines and their association with the three endometriosis subphenotypes. This allowed us to assemble cytokine signatures, which revealed dysregulated biological processes associated with the pathogenesis/pathophysiology of the three endometriosis subphenotypes. These results might direct new endometriosis therapeutic and diagnostic efforts toward subphenotype clinically relevant cytokine targets.

## 2. Results

### 2.1. Peritoneal Fluid Cytokines Inadequately Describe Endometriosis Stages and Their Heterogeneity

PF samples from 132 women (without endometriosis, EM− = 59; with endometriosis, EM+ = 73) (Table 1) were analyzed for 48 cytokines, using a validated multiplex immunoassay platform. Among the cytokines analyzed, 38 (79%) cytokines were detected and compared using univariate analysis (Appendix A). Because inflammatory cascades are brought about by the interaction of pro-inflammatory and anti-inflammatory cytokines [21], multivariate analysis such as PLSR provides a powerful way to obtain cytokine signatures to molecularly distinguish the patient groups in relation to disease presence and stages. Using partial least squares regression (PLSR), we identified a cytokine signature comprising IFN-α2, IL-12p70, IL-18, SCGF-β, VEGF-A, IL-3, and HGF that distinguished EM- from EM+ (Figure 1A). Notably, IFN-α2, IL-12p70, IL-18, SCGF-β, and VEGF-A are cytokines previously identified to be significantly altered between EM- from EM+ individuals [29,30,31]. However, a significant overlap between the EM- and EM+ clusters was observed (cumulative principal component (PC) score: 43%; Figure 1B), congruent with other studies [32]. We used the cumulative PC score as a measure of how well the cytokines could distinguish between EM-, EM+, EM+_Mild_, and EM+_Severe_ patients (perfect separation of patient clusters at 100% cumulative PC score). When EM+_Mild_ and EM+_Severe_ individuals were compared to EM-, the PLSR separations improved slightly, as indicated by higher cumulative PC scores of 51% and 49% (Figure 1C and Appendix A). PLSR modeling of PF cytokines in EM+_Mild_ from EM+_Severe_ separated at 59%, but remains relatively poor as observed in the other models of disease stages (EM- versus EM+_Mild_, EM- versus EM+_Severe_), suggesting heterogeneity of disease stages and subsequent incompleteness to distinguish endometriosis (Appendix A).

### 2.2. Peritoneal Fluid Cytokine Signatures Delineate Endometriosis Subphenotypes

The three subphenotypes of endometriosis, OE, PE, and DIE, present peculiar clinicopathological characteristics, which prompted us to hypothesize that PF cytokines might describe endometriosis better, viz-a-viz capturing molecular variations of the subphenotypes. Based on PF cytokines, PLSR modeling of endometriosis subphenotypes resulted in better models than disease stages (cumulative PC scores: 77% to 92%, *p* < 0.0001; Figure 1D and Figure 2A–C; Appendix A). PLSR scores plots revealed clear delineation of OE, PE, and DIE (Appendix A), suggesting their distinct molecular makeups. In addition, analysis of the subphenotypes with controls showed significant moderate separation (cumulative PC score = 55–69%, *p* < 0.001; Appendix A).

DIE has been considered a specific entity in which lesions penetrate more than 5 mm underneath the peritoneum [9,33]. Indeed, PC scores from comparisons of PE with OE (Figure 2A) was smaller than PC scores comparing PE or OE with DIE (Figure 2B,C), suggesting greater distinction of DIE from the superficial subphenotypes. A six-cytokine signature comprising IL-1α, IL-7, IL-8, MCP-1, MIF, and TNF-α distinguished OE from PE was identified (Figure 2D). Comparing OE to DIE, a seven-cytokine signature comprising IL-1α, IL-1RA, IL-8, IL-12p40, IL-12p70, IL-16, and TNF-α was identified (Figure 2E). Comparing DIE to PE, an all-upregulated six-cytokine signature of IL-8, IL-12p70, IL-16, IL-18, MCP-1, and MIP-1α that correlated to DIE was identified (Figure 2F). Cross-correlation and hierarchical clustering of cytokines showed not only intercorrelated inflammatory cytokines (e.g., TNFα, IL-1β, IL-10, and IL-1RA), but also cytokines that are anti-correlated (e.g., IL-12p40 and IL-12p70) [34], affirming the underlying biological information embedded within the PLSR-derived cytokine signatures (Figure 3. Appendix A shows the univariate statistical analysis.

### 2.3. Subphenotype Cytokine Signatures Are Associated with Different Biological Processes.

To examine whether the cytokine signatures associated with different endometriosis subphenotypes showed themes of functional categorization, each signature was investigated by using the gene ontology (GO) classification system (Table 2). In OE relative to PE, enriched biological processes include the following: negative regulation of extrinsic apoptotic signaling pathway; positive regulation of B-cell proliferation; positive regulation of angiogenesis; and positive regulation of ERK1 and ERK2 cascade. In OE relative to DIE, natural-killer-cell-mediated cytotoxicity is directed against tumor-cell target, cell-cycle arrest, and cell migration, and tyrosine phosphorylation of Stat4 protein. Smooth muscle cell metaplasia that is associated with DIE was also identified in our pathway enrichment analysis [35], providing assurance that our cytokine-signature-derived identified meaningful subphenotype biological characteristics. In PE versus DIE, enriched biological processes include the following: positive regulation of interferon-gamma production, protein kinase B (also known as AKT) signaling, and MAPK cascade.

## 3. Discussion

Reports have long recognized endometriotic lesion heterogeneity in characteristics such as color, size, and location [3,4,5]. There is increasing evidence that there is marked variation in lesion subphenotypes at the molecular level [36]. Santulli et al. reported that DIE is associated with higher PF advanced oxidation protein products than PE and OE [37]. Immunohistochemical analyses showed a higher prevalence of cyclooxygenase-2 protein expression in OE (78.5%) compared to 11.1% and 13.3% in PE and DIE [38], whereas nerve growth factor is expressed higher in DIE compared to PE and OE [39]. Methylation promotor sites in ectopic lesions are potentially different between the subphenotypes [40]. These reports, collectively, are in strong agreement with our results shown herein, in that OE, PE, and DIE are associated with unique cytokine signatures and corresponding enriched biological pathways.

Our study is likely to have implications for endometriosis therapeutics and biomarker discovery, particularly in the era of targeted treatment and precision medicine [41]. Further interpretation of our observations raises the potentiality of different therapies needed for treating patients with different endometriosis subphenotypes. It is apparent that broad, untargeted usage of therapeutics does not bring about clear clinical benefit across all endometriosis patients [2]. Endometriosis subphenotypes may partly explain these clinical phenomena. While targeted therapies have not been investigated in endometriosis, data from our study offer the possibility of exploiting each subphenotype’s unique dependence on critical non-hormonal signaling or inflammation pathways to tailor therapy. Drug side-effect profile and efficacy are two main considerations of treatment choices [42]. Our findings, along with those of others, warrant further deep characterization of cytokine profiles and targeted use of immunomodulators, given the centrality of cytokines cells in the process of establishment and maintenance of this peritoneal disease. Due to often described elevation of peritoneal TNFα in endometriosis by us and others, blockers of TNFα, such as etanercept, leflunomide, and infliximab, were evaluated in preclinical models and in small randomized clinical trials [43,44,45,46]. Other immunomodulators, such as levamisole, and IL-12 inhibitors, have also been investigated in experimental models of endometriosis, with certain success [47]. In approved drugs such as infliximab, levamisole, and leflunomide, where safety profiles are established, monotherapeutic efficacy in humans remains untested or uncertain. While larger and more clinical trials are certainly required to test drug efficacy, immunomodulators are not always effective alone and require further development, along with new combination strategies, in order to enhance their efficacy, such as the use of combinatorial therapy or immunotherapy. These combination therapies can be classified based on their strategic targets: firstly, blockage of multiple cytokines based on the abovementioned signatures; secondly, to promote T- or NK-cell priming by enhancing lesion-associated antigen presentation; and thirdly, to target the immunosuppressive environment [48,49]. In the first strategy, choosing the key cytokines is critical, as cytokines are also required physiologically for endometrial growth, decidualization, and implantation [50]. In the second and third strategies, deep phenotyping of peritoneal immune cells and lesion microenvironments in endometriosis is needed before [51,52]. These strategies and their corresponding challenges must be overcome before the promise of an efficacious immunomodulatory therapy can be realized. Our study also has clinical implications on the biomarkers of endometriosis and biomarkers that are predictive of response to treatment. Heterogeneity in endometriosis has had a negative impact on the discovery and validation of predictive biomarkers, insofar that the lack of robust biomarkers is a critical area the community is working toward surmounting [53,54,55]. While this is not addressed in our study, our study creates the background for future investigations of stratifying patients according to their subphenotypes, prior to investigation of biomarkers.

During endometriosis development, cytokines develop a reciprocal interaction between the lesions and their surroundings, which constitute the modular microenvironment, influencing the evolution of lesion subphenotypes [14,20,56]. In this study, we opted to study the PF as a reflection of the spatial heterogeneity in endometriosis subphenotypes. By doing so, we were able to investigate the dynamic crosstalk between the secreted cytokines of lesions and their juxtaposed microenvironments, and their reciprocal adaptations. Indeed, ovarian endometriomas by growing near or at ovarian tissues with elevated CYP19A1 expressions are exposed to estrogen levels higher than that experienced by lesions implanted in the peritoneum or deep areas [57], creating an ovarian-specific microenvironment. Similarly, the transcriptome of the peritoneum in EM+ women is fundamentally different from that of EM- women [58,59,60]. The different expressions of inflammatory and adhesion molecules such as ICAM-1, matrix metalloproteases, and IL-6 confer a preferentially adhesive microenvironment for lesion implantation [61]. Elevated peritoneal reactive oxygen species acting through various signaling pathways, such as MAPK, ERK, and AKT, regulate gene expression of cytokines and cell adhesion molecules, which create microenvironments that promote various aspects of endometriosis development and its deleterious effects [62,63,64]. In DIE, significant fibrotic accumulation and smooth muscle differentiation secondary to the proliferation of stromal cells and inflammation are commonly noted and present yet another kind of microenvironment plausibly enhancing chronic inflammation [65,66]. In addition, the lesion microenvironment and its secreted cytokines plausibly play a key role in therapeutic approach of endometriosis, noting the elicitation of drug resistance by stromal microenvironment surrounding tumors in cancer [67]. Transcriptomic studies, such as those by Suryawanshi et al. [68], would be especially illuminating, had an analysis of the three endometriosis subphenotypes been performed in terms of elucidating each subphenotype’s underlying pathogenic/pathophysiological mechanisms and potential “druggability”. While additional experiments comparing transcriptome, epigenome, and proteome in PE, OE, and DIE are needed, our results suggest heterogeneity in the subphenotypes, at least in their cytokine levels, is associated with their microenvironments.

We used the GO database to analyze the cytokine signatures for enriched biological processes pertaining to the subphenotypes. Several key themes emerged: OE cytokines centered on proliferation/apoptosis regulation linked to ERK1 and ERK2 signaling, compared to PE; PE cytokines were implicated in MAPK and AKT signaling and highly inflammatory environments, compared to DIE; DIE cytokines were associated with cytotoxicity directed against “tumor cell target” and smooth muscle cell metaplasia, consistent with known outcomes associated with DIE [35]. Further, our analysis identified STAT4 regulation, and so it is congruent with studies by Zamani et al. and Bianco et al., showing *STAT4* polymorphism and susceptibility to endometriosis [69,70]. Increased angiogenesis is linked to the development of maintenance of endometriotic lesions, and our analysis suggests stronger angiogenesis in ovarian endometriomas [71,72], plausibly due to the strong induction of angiogenesis under high levels of ovarian estrogen. The GO analysis also revealed perturbations in immune cells. The positive regulation of natural killer cell activation could be construed as a compensatory mechanism consistent with reduced natural killer cell cytotoxicity [73], facilitating survival of regurgitated menstrual tissues at ectopic sites, probably more so in OE than DIE, as suggested. The combination of reduced IL-17 production and T-cell proliferation suggests exacerbated T-regulatory cells or Tregs activity [74].

Strengths of this study include the use of a carefully phenotyped clinical study population, use of a large, unbiased multiplexed cytokine approach, and advanced biostatistics. Additionally, this study, which was conducted in Singapore, represents a unique strength, as the study population pertains to Asians (Chinese, Malays, Indians, and Filipinos), providing a defined patient background for educated comparison and generalizability of the results when required. Limitations of this study include its observational nature, lack of longitudinal data, and the difficulty in dissecting the specific roles of cytokines within the molecular signatures. It is recognized that findings from this study are preliminary and will need to be validated in other populations, given that cytokine signatures may differ in other study populations. The cellular and molecular mechanisms of endometriosis development are likely to be overlapping (as observed in IL-8) and manifold, and many cytokines are able to induce the pathways. Thus, it is likely that multiple inflammatory pathways induced by a variety of stimuli might lead to endometriosis subphenotype development and endocrine failure. Further experiments will be necessary to define the precise roles of cytokines in the immune regulation of endometriosis. Taken together, the clustering of cytokines into functional groups hints at different pathogenic/pathophysiological mechanisms defining endometriosis subphenotypes. This would have important clinical ramifications, with the prediction that the endometriosis subphenotypes might require different treatment strategies and meet the need of a more personalized approach for endometriosis management [75].

## 4. Materials and Methods

### 4.1. Subjects and Sample Collection

Peritoneal fluids (PF) were collected from women participants (*n* = 132), comprising 59 women who are endometriosis-free (EM-) and 73 with endometriosis (EM+) undergoing laparoscopic procedures for suspected endometriosis, infertility, sterilization procedures, and/or pelvic pain recruited in KK Women’s and Children’s Hospital, Singapore, and Singapore General Hospital, Singapore, under Centralized Institutional Research Board approval (CIRB 2010-167-D, approved 15 January 2016). Diagnostic laparoscopy was performed on all patients, with careful inspection of the uterus, fallopian tubes, ovaries, pouch of Douglas, and the pelvic peritoneum by gynecologists subspecializing in reproductive endocrinology and infertility. PFs were prepared as previously described [18], in line with Endometriosis Phenome and Biobanking Harmonisation Project Standard Operating Procedures [76]. Presence of endometriosis was systematically recorded and scored according to the revised American Fertility Society classification (rAFS) of endometriosis [77,78]. All patients gave written informed consent. Exclusion criteria included menstruating patients, post-menopausal patients, patients on hormonal therapy (e.g., norethisterone and microgynon) for at least three months before laparoscopy, and other confounding diseases, such as diabetes, adenomyosis, or any other chronic inflammatory diseases (rheumatoid arthritis, inflammatory bowel disease, systemic sclerosis, etc.). Patient characteristics are shown in Table 1. For the subjects undergoing robot-assisted excision of DIE at Singapore General Hospital, all women were preoperatively assessed for pelvic pain, bleeding, or fertility problems by trans-vaginal or trans-rectal ultrasound (if *virgo intacta*), using Voluson ultrasound machines (GE Healthcare). All sonographers were trained in the identification of DIE. All cases had pelvic DIE lesions of >5 mm, as well as evidence of partial or complete obliteration of the POD. In our unit, POD obliteration was taken to be the evidence of bowel involvement, and the use of robot-assisted technology for surgical excision was triggered due to suspected increased surgical complexity. Robot-assisted excision was performed by using the da Vinci Surgical Si™ system (Intuitive Surgical). The possible overlapping of the three lesion subphenotypes led us to classify the patients according to the worst lesion found in each subject, based on endometriosis subphenotype grouping by Chapron et al. [66,79]. Subjects who did not have endometriosis or have benign gynecological presentations, such as uterine fibroids and benign ovarian cysts, were taken as the endometriosis-free control group (EM-). All PFs were stored at −80 °C, until further analysis.

### 4.2. Multiplex Immunoassay Analysis

Cytokines were detected and measured by using multiplex immunoassay (BioRad, Hercules, CA, USA; Appendix A), as previously described [50]. Briefly, 10 μL of PF was mixed with 10 μL of primary antibody-conjugated magnetic beads on in a 96 DropArray plate (Curiox Biosystems, Singapore) and rotated at 450 rpm on a plate shaker for 120 min at 25 °C, while protected from light. Subsequently, the plate was washed three times with wash buffer on the LT210 Washing Station (Curiox), before adding 5 μL of secondary antibody and rotating at 450 rpm for 30 min at 25 °C, protected from light. The plate was washed three times with wash buffer, and 10 μL of streptavidin-phycoerythrin was added and rotated at 450 rpm for 30 min at 25 °C, protected from light. The plate was again washed thrice with wash buffer. Then, 60 μL of reading buffer was added and transferred to a 96-conical-well microtiter plate, and the samples were read by using the Bio-Plex Luminex 200 (BioRad). The beads are classified by the red classification laser (635 nm) into its distinct sets, while a green reporter laser (532 nm) excites the phycoerythrin, a fluorescent reporter tag bound to the detection antibody. Quantitation of the 48 cytokines in each sample was then determined by extrapolation to a six- or seven-point standard curve, using five-parameter logistic regression modeling. Assay CV averaged <12%. When samples were detected in less than 50% of patients or below the lower limit of quantitation, they were considered undetected. Calibrations and validations were performed prior to runs and on a monthly basis, respectively.

### 4.3. Sample Size Calculation

We estimated the sample size before commencing the study: We assumed a 95% sensitivity of the cytokine signatures in distinguishing the subphenotypes, and a 95% confidence interval (CI) of approximately ±10.0% would need 18 cases of one endometriosis subphenotype and the same number of another subphenotype. This statistical analysis, assuming the sensitivity of the signature cutoff is *p* = # test positive/*N*, where the sample size (*n*), number of women diagnosed with a particular endometriosis subtype (DIE, OE, PE), necessary to estimate *p* with precision ± L is given by the formula n = Z*Z*p*(1-*p*)/(*L***L*), where Z corresponds to the correct percentile of the standard normal distribution [80]. Our clinical experience noted a 1:2:1 PE:OE:DIE ratio, and hence we targeted 36 OE cases, and the patient numbers were rounded to fall within the estimated sensitivity.

### 4.4. Statistical Analysis

GraphPad Prism 6 (GraphPad Software Inc., San Diego, CA, USA) was used for statistical analyses. Data were checked for normal distribution, using the Kolmogorov–Smirnov test. Unpaired or paired Student’s *t*-test was performed, as appropriate, to determine statistical significance between groups for normally distributed data. Mann–Whitney U test was used for non-normally distributed data. For comparing three or more groups, the data were analyzed by using one-way ANOVA, followed by the Student’s *t*-test with Bonferroni adjustment for pairwise comparisons. A *p-*value <0.05 was deemed to be statistically significant. A two-tailed Pearson correlation matrix was first performed with a confidence interval of 95%, followed by hierarchical clustering and the results plotted as heatmap 1.0 (GPS HemI 1.0 Heatmap Illustrator), using the clustering method of Average linkage and similarity metric of Pearson distance. Cytokines were further analyzed and signatures obtained by partial least squares regression (PLSR) modeling (Unscrambler X version 10.1) after the normalization of data by performing log2 transformation. Full cross-validation was applied in PLSR to increase model performance and for the calculation of coefficient regression values. PLSR models were compared by using Q-residuals obtained from the best principal components.

### 4.5. Pathway Enrichment Analysis

PLSR coefficient-derived cytokines were imputed into the Database for Annotation, Visualization and Integrated Discovery (DAVID) and cross-referenced against Gene Ontology (GO), for biological-process-enrichment analysis [81]. The *p*-values <0.05, based on Fisher Exact analysis and fold change >10 were imposed as enriched pathways. Bonferroni and Benjamini adjusted *p-*values are reported in Appendix A. As a result of these corrections, the adjusted *p*-values get larger, and it could hurt the sensitivity of discovery if overemphasizing them [81,82]. Furthermore, these commonly used multiple testing-adjustment methods assume independence of tests, which in cytokines studies translates to a questionable assumption that all cytokines operate independently; instead, cytokines form biological networks [21,22].

## 5. Conclusions

Taken together, this study suggests that women with endometriosis subphenotypes can be stratified molecularly. Although the determination of lesion heterogeneity has not yet formed part of the clinical decision-making process in endometriosis, subphenotype-specific cytokine signatures suggest that suitable targeted therapeutic approaches specific to each individual patient’s lesion and peritoneal heterogeneity composition may be tailored, and also the consideration of endometriosis subphenotypes for biomarker discovery studies. We anticipate our results to stimulate further studies, with accompanying detailed anatomical locations, and the use of genomic, proteomic, and metabolomic studies to evaluate the subphenotype molecular characteristics.

## Figures and Tables

**Figure 1 ijms-21-03515-f001:**
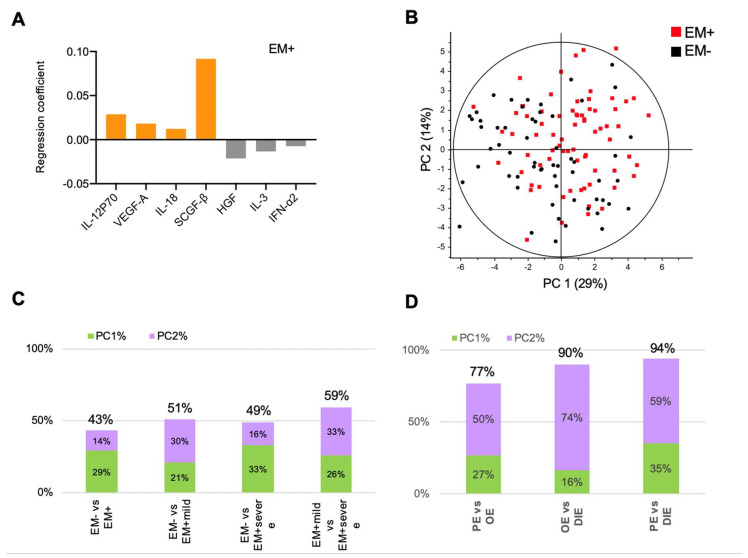
Peritoneal fluid cytokines associate with endometriosis stages. (**A**) Partial least squares regression (PLSR) coefficient analysis revealed a signature comprising elevated IL-12p70, IL-18, VEGF-A, and SCGF-β and decreased IFN-α2, IL-3, and HGF that distinguished women with endometriosis (EM+) from women without (EM-). (**B**) Modeling by PLSR scores plot reveals overlap of EM- and EM+, suggesting heterogeneity in the peritoneal fluid environment. PLSR-derived principal component scores of principal component 1 (PC1) and principal component 2 (PC2) of (**C**) stages and (**D**) subphenotypes. Cumulative principal-component scores are shown at the top of each bar.

**Figure 2 ijms-21-03515-f002:**
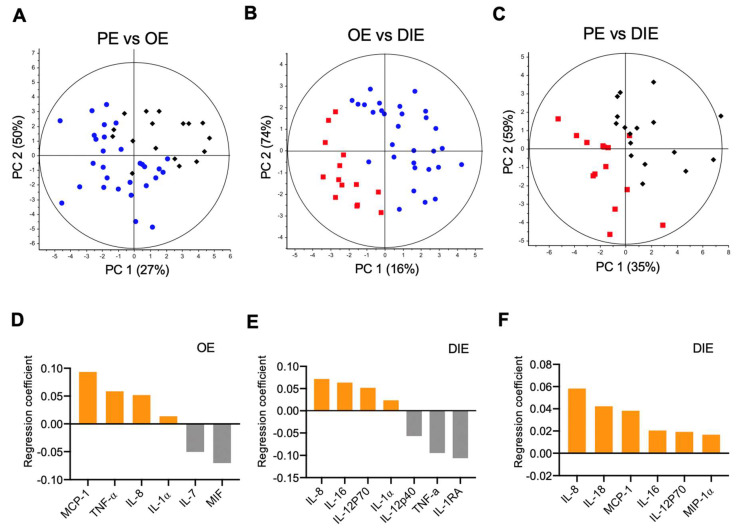
Peritoneal fluid cytokines show distinct delineation of endometriosis subphenotypes. Partial least squares regression (PLSR) models separated (**A**) ovarian endometriomas from peritoneal endometriosis, (**B**) ovarian endometriomas from deep infiltrating endometriosis, and (**C**) peritoneal endometriosis from deep infiltrating endometriosis. The principal component (PC) scores show good separation of endometriosis subphenotypes by using PF cytokines. (**D**–**F**). Corresponding PLSR coefficient analyses reveal cytokine signatures delineating the various subphenotypes. Elevated cytokines associated with a particular endometriosis subphenotype (OE, PE, or DIE) relative to its comparator appear in the same upper or lower half of the plot.

**Figure 3 ijms-21-03515-f003:**
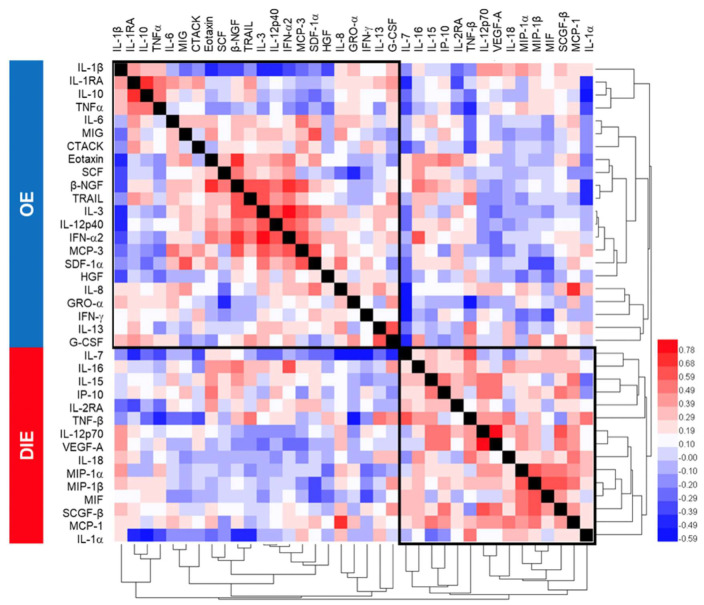
Correlation matrix of peritoneal fluid cytokines. Hierarchical clustering was performed on Spearman *r*-values between the subphenotypes ovarian endometriosis, and deep infiltrating endometriosis revealed consistency in PLSR-derived cytokine signatures that segregated OE from DIE.

**Table 1 ijms-21-03515-t001:** Patient characteristics.

Characteristics	EM- (*n* = 59)	EM+ (*n* = 73)	*p*-Value ^†^
**Age, *y***			0.844
Mean	35	35	
Range	22–51	25–45	
**ASRM Stage**			
I–II	NA	31	
III–IV	NA	42	
Subtype			
Peritoneal	NA	17	
Endometrioma	NA	30	
Deep infiltrating	NA	14	
Undetermined	NA	12	
**Preoperative pain symptoms** *^a^*			<0.0001
Dysmenorrhea	20	56	
Dyspareunia *^b^*	5	20	
**Menstrual phase**			0.290
Proliferative	27	41	
Secretory	32	32	
**Race**			0.650
Chinese	36	46	
Malay	14	13	
Others *^c^*	9	14	

^†^ Student’s *t*-test was performed for age, and Chi square for categorical data in pain, menstrual phase, and race. ^a^ 18 patients present with both dysmenorrhea and dyspareunia. ^b^ 10 patients did not have coitus at time of surgery. ^c^ Includes Indians and Filipinos

**Table 2 ijms-21-03515-t002:** Functional enrichment analysis of endometriosis subphenotype cytokine signatures.

(A) OE vs. PE		
Term Identifier	Fold Enrichment	*p*-Value
immune response	29.9	1.90 × 10^−3^
positive regulation of B-cell proliferation	215.3	7.00 × 10^−3^
inflammatory response	33.2	1.50 × 10^−3^
negative regulation of extrinsic apoptotic signaling pathway in absence of ligand	226.9	6.60 × 10^−3^
positive regulation of angiogenesis	73	2.00 × 10^−2^
positive regulation of ERK1 and ERK2 cascade	48	3.10 × 10^−2^
cell proliferation	22.9	6.40 × 10^−2^
negative regulation of apoptotic process	18.5	7.90 × 10^−2^
**(B) OE vs. DIE**		
Term Identifier	Fold Enrichment	*p*-Value
immune response	31.9	6.10 × 10^−5^
positive regulation of natural killer cell activation	1343.4	1.20 × 10^−3^
positive regulation of NK T-cell activation	1343.4	1.20 × 10^−3^
positive regulation of tyrosine phosphorylation of Stat4 protein	1679.2	9.50 × 10^−4^
positive regulation of lymphocyte proliferation	959.5	1.70 × 10^−3^
positive regulation of natural killer cell mediated cytotoxicity directed against tumor cell target	959.5	1.70 × 10^−3^
positive regulation of mononuclear cell proliferation	2238.9	7.10 × 10^−4^
response to UV-B	746.3	2.10 × 10^−3^
positive regulation of smooth muscle cell apoptotic process	746.3	2.10 × 10^−3^
negative regulation of interleukin-17 production	610.6	2.60 × 10^−3^
positive regulation of T-cell-mediated cytotoxicity	516.7	3.10 × 10^−3^
defense response to protozoan	353.5	4.50 × 10^−3^
negative regulation of smooth muscle cell proliferation	231.6	6.90 × 10^−3^
positive regulation of interferon-gamma production	146	1.10 × 10^−2^
positive regulation of cell adhesion	156.2	1.00 × 10^−2^
positive regulation of T-cell proliferation	111.9	1.40 × 10^−2^
cellular response to lipopolysaccharide	59.4	2.70 × 10^−2^
cytokine-mediated signaling pathway	51.3	3.10 × 10^−2^
cell cycle arrest	47.6	3.30 × 10^−2^
cell migration	39.1	4.00 × 10^−2^
**(C) PE vs. DIE**		
Term Identifier	Fold Enrichment	*p*-Value
immune response	39.9	1.60 × 10^−5^
positive regulation of protein kinase B signaling	100	1.50 × 10^−2^
positive regulation of inflammatory response	115	1.30 × 10^−2^
cellular response to organic cyclic compound	142.3	1.10 × 10^−2^
positive regulation of interferon-gamma production	182.5	8.20 × 10^−3^
lipopolysaccharide-mediated signaling pathway	262.4	5.70 × 10^−3^
MAPK cascade	32	4.60 × 10^−2^
cell–cell signaling	33.1	4.50 × 10^−2^
inflammatory response	22.2	6.60 × 10^−2^

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
