# Peer review of "Peritoneal Fluid Cytokines Reveal New Insights of Endometriosis Subphenotypes"

_ijms, 2020, doi:10.3390/ijms21103515_

Round 1

Reviewer 1 Report

The paper presented for the review aims to investigate peritoneal fluid cytokines and their association with the three endometriosis subphenotypes. This study is well-written and easy to follow. However, there are some remarks that have to be addressed:

  • Did the authors calculate the sample size before starting their study? Otherwise, the authors should perform a post-hoc calculation of the power of their study, in order to ensure adequate significance of their results. Indeed, the large sample size is not a guarantee of an adequate power of the study.
  • The authors did not discuss strengths and limitations of their study. I recommend adding this point in the Manuscript.
  • As correctly underlined by the Authors, immune cells, adhesion molecules, extracellular matrix metalloproteinase and pro-inflammatory cytokines play an active role in the pathogenesis of endometriosis. I would discuss these points in the light of new theories about the pathogenesis of endometriosis, referring to: PMID: 29743986; PMID: 29057034.

Author Response

All revisions in the revised manuscript are highlighted in yellow.

Thanks.

Reviewer 1

The paper presented for the review aims to investigate peritoneal fluid cytokines and their association with the three endometriosis subphenotypes. This study is well-written and easy to follow. However, there are some remarks that have to be addressed:

  • Did the authors calculate the sample size before starting their study? Otherwise, the authors should perform a post-hoc calculation of the power of their study, in order to ensure adequate significance of their results. Indeed, the large sample size is not a guarantee of an adequate power of the study.

Authors’ response: We estimated the sample size before commencing the study:  we assumed a 95% sensitivity of the cytokine signatures in distinguishing the subphenotypes, and a 95% confidence interval (CI) of approximately ±10.0% would need 18 cases of one endometriosis subphenotype and the same number of another subphenotype. This statistical analysis, assuming the sensitivity of the signature cutoff is p = # test positive/N, where the sample size (N), number of women diagnosed with a particular endometriosis subtype (DIE, OE, PE), necessary to estimate p with precision ± L is given by the formula n = Z*Zp(1-p)/(L*L), where Z corresponds to the correct percentile of the standard normal distribution1. Our clinical experience noted a 1:2:1 PE:OE:DIE ratio, and hence we targeted 36 OE cases and the patient numbers were rounded to fall within the estimated sensitivity. We have added this paragraph to the Methods section of the manuscript.

1W.G. Cochran, Methodological problems in the study of human populations, Ann. N. Y. Acad. Sci. 107 (1963) 476–489

  • The authors did not discuss strengths and limitations of their study. I recommend adding this point in the Manuscript.

Authors’ response: We have added a discussion of strengths and limitations of this study at lines 279-292 as suggested.

  • As correctly underlined by the Authors, immune cells, adhesion molecules, extracellular matrix metalloproteinase and pro-inflammatory cytokines play an active role in the pathogenesis of endometriosis. I would discuss these points in the light of new theories about the pathogenesis of endometriosis, referring to: PMID: 29743986; PMID: 29057034.

Authors’ response: We thank the reviewers for the suggested literatures, and we have incorporated reactive oxygen species as plausible key players in endometriosis development at lines 248-251.

Reviewer 2 Report

I revised the manuscript entitled “Peritoneal fluid cytokines reveal new insights of endometriosis sub phenotypes” (Manuscript Number: ijms-793942).

The topic of this manuscript falls within the scope of the International Journal of Molecular Sciences.

I was particularly pleased to review this paper. In my honest opinion, the topic is interesting enough to attract the readers’ attention. Methodology is accurate and conclusions are supported by the data analysis. Nevertheless, authors should clarify some point and improve the discussion citing relevant and novel key articles about the topic and discussion limitations of the study that are not enough evidenced in the discussion.

In general, the Manuscript may benefit from several major revisions, as suggested below:

- All the text needs a language revision, in order to improve some typos and grammatical errors.

  • Abstract. I would suggest clarifying the term “disease severity”. Moreover, I would suggest reporting more details about the investigated population, in which women cytokines were investigated and how endometriosis was diagnosed and classified. Moreover. I would suggest better reporting results. Which pattern of cytokines are associated to which subtype of endometriosis is unclear.
  • - Introduction. Lines 65-71 should be better clarified. I would suggest better discussing available theories about endometriosis pathogenesis referring to PMID: 31717614 and PMID: 32046116.
  • - Methods. I would suggest improving the description of study population with inclusion and exclusion criteria adopted to select eligible women, i.e. were ongoing therapy allowed?, many information are lacking to allow the selection of a comparable population. Moreover, the study results apply to the investigated population and details about included women are required to understand the generalizability of the study results.
  • - Methods. I would suggest clearly stating the definitions adopted for the three type of endometriosis. Moreover, what does it mean “according to the worst lesion”? Which was the adopted definition? This is not objective but subjective, the adopted criteria need to be reported.
  • - I would suggest checking the use of abbreviation at the first use in the manuscript and abstract.
  • - I would suggest reporting how and who assessed the endometriosis severity, if it refers to the endometriosis stage, I would suggest using stage instead of severity.
  • - Were the p-values reported in table 2 already adapted with Bonferroni method? More details should be reported about how many comparisons were adopted to correct the p values.
    - Discussion. Regarding the discussion of therapeutic options, I would suggest better discussing the pros and cons of the use of cytokines as target of therapies and the state of the art about this strategy to show the complexity to achieve such a result.
  • - Details about patient consent and IRB approval are not reported.

Author Response

All revisions in the revised manuscript are highlighted in yellow.

Thanks.

Reviewer #2

The topic of this manuscript falls within the scope of the International Journal of Molecular Sciences.

I was particularly pleased to review this paper. In my honest opinion, the topic is interesting enough to attract the readers’ attention. Methodology is accurate and conclusions are supported by the data analysis. Nevertheless, authors should clarify some point and improve the discussion citing relevant and novel key articles about the topic and discussion limitations of the study that are not enough evidenced in the discussion.

In general, the Manuscript may benefit from several major revisions, as suggested below:

- All the text needs a language revision, in order to improve some typos and grammatical errors.

  • I would suggest clarifying the term “disease severity”. Moreover, I would suggest reporting more details about the investigated population, in which women cytokines were investigated and how endometriosis was diagnosed and classified. Moreover. I would suggest better reporting results. Which pattern of cytokines are associated to which subtype of endometriosis is unclear.

Authors’ response: We thank the reviewer for the comments on improving the abstract. Disease severity has been changed to disease stages throughout the text, and more information of patient diagnosis and results have been provided within the imposed abstract word limit of 200 words.

  • Lines 65-71 should be better clarified. I would suggest better discussing available theories about endometriosis pathogenesis referring to PMID: 31717614 and PMID: 32046116.

Authors’ response: The paragraph has been re-written to introduce the various theories and hypotheses as lines 64-80.

  • I would suggest improving the description of study population with inclusion and exclusion criteria adopted to select eligible women, i.e. were ongoing therapy allowed?, many information are lacking to allow the selection of a comparable population. Moreover, the study results apply to the investigated population and details about included women are required to understand the generalizability of the study results.

Authors’ response: We appreciate the reviewer for highlighting this important point which was included in reference 18. However, given the need for clarity within the study itself, inclusive/exclusion criteria have been included. They are written as such: “Diagnostic laparoscopy was performed on all patients, with careful inspection of the uterus, fallopian tubes, ovaries, pouch of Douglas and the pelvic peritoneum by gynaecologists subspecializing in reproductive endocrinology and infertility. PFs were previously described, in line with Endometriosis Phenome and Biobanking Harmonisation Project Standard Operating Procedures. Presence of endometriosis was systematically recorded and scored according to the revised American Fertility Society/ American Society for Reproductive Medicinewww classification (rAFS/ASRM) of endometriosis. All patients gave written informed consent. Exclusion criteria included menstruating patients, post-menopausal patients, patients on hormonal therapy (e.g. norethisterone, microgynon) for at least three months before laparoscopy, and other confounding diseases such as diabetes, adenomyosis or any other chronic inflammatory diseases (rheumatoid arthritis, inflammatory bowel disease, systemic sclerosis etc).”

  • I would suggest clearly stating the definitions adopted for the three type of endometriosis. Moreover, what does it mean “according to the worst lesion”? Which was the adopted definition? This is not objective but subjective, the adopted criteria need to be reported.

Authors’ response: According to the protocol used in the department, descriptions of the anatomical pelvic situation, scored according to rAFS/ASRM, and sketches of the endometriotic lesion implantation sites drawn. PE was defined by the presence of endometriosis at pelvic wall, Douglas cul-de-sac, or uterine surface. Endometriomas was defined by the presence of histologically confirmed endometriosis at the ovaries. DIE was defined as the presence of histologically confirmed endometriosis infiltrating to a depth of at least 5 mm beneath the peritoneal surface. Cases were discarded if written and graphic description did not coincide. In cases where there is overlap of the three types of lesions rAFS/ASRM staging was first applied followed by the dominant lesion found in each subject which is based on highest scoring lesion on the rAFS scale.

  • I would suggest checking the use of abbreviation at the first use in the manuscript and abstract.

Authors’ response: We thank the reviewer again for his/her meticulous review. The cytokines names, PLSR, EM-, EM+ and GO have been spelled out in the first instance.

  • I would suggest reporting how and who assessed the endometriosis severity, if it refers to the endometriosis stage, I would suggest using stage instead of severity.

Authors’ response: As addressed in the earlier comment, we have noted that “Diagnostic laparoscopy was performed on all patients, with careful inspection of the uterus, fallopian tubes, ovaries, pouch of Douglas and the pelvic peritoneum by gynaecologists subspecializing in reproductive endocrinology and infertility.”  Also as above, the word “disease stage” has been used in place of severity throughout the text.

  • Were the p-values reported in table 2 already adapted with Bonferroni method? More details should be reported about how many comparisons were adopted to correct the p values.

Authors’ response:  The reported p-values were obtained from a modified Fisher Exact statistical test. The modified Fisher Exact statistics is used to calculate based on corresponding DAVID gene IDs by which all redundancies in original IDs are removed. The modified Fisher Exact p-value, for gene-enrichment analysis, ranging from 0 to 1 and a p-value is equal or smaller than 0.05 to be considered strongly enriched in the annotation categories. For our internal understanding, we performed Bonferroni and Benjamini corrections (Table S5). However, we opted not to report as the corrections are known to be one of the most conservative approaches. As a result of corrections, the adjusted p-values get larger and, it could hurt the sensitivity of discovery if overemphasizing them1,2.  Furthermore, these commonly used multiple testing adjustment methods assume independence of tests, which in cytokines studies translates to a questionable assumption that all cytokines operate independently; instead, cytokines form networks3,4.

1Nature Protocols 2009, Vol. 4, 44–57
2The Annals of Statistics 2001, Vol. 29, No. 4, 1165–1188
3Biochim. Biophys. Acta - Mol. Cell Res., 2011, Vol. 1813, 2165–2175
4Cytokine, 2013, Vol. 62, 165–173

  • Regarding the discussion of therapeutic options, I would suggest better discussing the pros and cons of the use of cytokines as target of therapies and the state of the art about this strategy to show the complexity to achieve such a result.

Authors’ response: We agree that this is an important point of discussion, and we appreciate this reviewer for giving us the opportunity to discuss this topic. We presented our discussion at Lines 209 -230.

  • Details about patient consent and IRB approval are not reported.

Authors’ response: We wish to highlight that the patient consent and IRB approval were reported in lines 300-314. In addition, we added a line on informed consent. Hence, the revised statement as follow: “Peritoneal fluids (PF) were collected from women (n=132), comprising of 59 women who are endometriosis-free (EM-) and 73 with endometriosis (EM+) undergoing laparoscopic procedures for suspected endometriosis, infertility, sterilization procedures and/or pelvic pain recruited in KK Women’s and Children’s Hospital, Singapore and Singapore General Hospital, Singapore under Centralized Institutional Research Board approval (CIRB 2010-167-D). All patients gave written informed consent”.

Round 2

Reviewer 2 Report

I revised the manuscript entitled “Peritoneal fluid cytokines reveal new insights of endometriosis sub phenotypes” (Manuscript Number: ijms-793942).

The topic of this manuscript falls within the scope of the International Journal of Molecular Sciences.

I was particularly pleased to review this paper. In my honest opinion, the topic is interesting enough to attract the readers’ attention. The methodology is accurate and conclusions are supported by the data analysis. Moreover, the authors addressed all the suggested revisions and I appreciated the manuscript improvement.